# Exploring the Nutritional Potential and Functionality of Hemp and Rapeseed Proteins: A Review on Unveiling Anti-Nutritional Factors, Bioactive Compounds, and Functional Attributes

**DOI:** 10.3390/plants13091195

**Published:** 2024-04-25

**Authors:** Marina Axentii, Georgiana Gabriela Codină

**Affiliations:** Faculty of Food Engineering, Ștefan cel Mare University of Suceava, 720229 Suceava, Romania; axentii.marina@outlook.com

**Keywords:** hemp, rapeseed, protein, functionality, anti-nutritional factors, bioactive compounds, food applications, digestibility, innovative products

## Abstract

Plant-based proteins, like those derived from hemp and rapeseed can contribute significantly to a balanced diet and meet human daily nutritional requirements by providing essential nutrients such as protein, fiber, vitamins, minerals, and antioxidants. According to numerous recent research papers, the consumption of plant-based proteins has been associated with numerous health benefits, including a reduced risk of chronic diseases such as heart disease, diabetes, and certain cancers. Plant-based diets are often lower in saturated fat and cholesterol and higher in fiber and phytonutrients, which can support overall health and well-being. Present research investigates the nutritional attributes, functional properties, and potential food applications of hemp and rapeseed protein for a potential use in new food-product development, with a certain focus on identifying anti-nutritional factors and bioactive compounds. Through comprehensive analysis, anti-nutritional factors and bioactive compounds were elucidated, shedding light on their impact on protein quality and digestibility. The study also delves into the functional properties of hemp and rapeseed protein, unveiling their versatility in various food applications. Insights from this research contribute to a deeper understanding of the nutritional value and functional potential of hemp and rapeseed protein, paving the way for their further utilization in innovative food products with enhanced nutritional value and notable health benefits.

## 1. Introduction

The escalating demand for supplementary products, especially among the young and elderly population [1,2], a surge in lactose intolerance and gluten sensitivity [3], a rise in the adoption of vegan or vegetarian diets [4,5], and an increasing demand for sustainable and environmentally friendly protein sources [6,7] are drivers that move plant-based protein products to a new level and make them clearly an important source of nutrients for the human diet. In the realm of food product development, conditioned by the exploration of alternative protein sources, the use of rapeseed and hemp proteins has gained significant traction, fueled by the increasing demand for sustainable and nutritious options, but also for relatively cheap alternatives for the human diet.

Among food trends and modern consumer requirements nowadays, hemp and rapeseed proteins stand out as promising contenders, offering plenty of nutritional benefits and functional properties conducive to innovative food formulations [8]. Hemp and rapeseed proteins are renowned for their complete amino acid profiles [9], boasting a rich array of essential nutrients for the human diet, overall health, and well-being. These plant-based proteins are not only abundant in protein content but also provide valuable dietary fiber, vitamins, minerals, and antioxidants, rendering them invaluable assets in promoting balanced diets and meeting daily nutritional requirements [10,11].

Of note, both hemp and rapeseed are oilseed crops, meaning they are grown in many parts of the world in order to produce biofuels, animal livestock, and oil for food use, although the amount of protein these sources provide among other oilseed crops should be taken into consideration as they can be successfully implemented into high-protein product recipes, with an enhanced texture and flavor [12,13]. According to the United States Department of Agriculture (USDA) data (Figure 1), the most protein-rich oilseed crop is soybean (36.4 g of protein per 100 g), followed by hemp (31.6 g of protein per 100 g) and mustard (26 g of protein per 100 g). In contrast, 100 g of rapeseed contains only 18.6 g of protein [14].

Beyond their nutritional prowess, hemp and rapeseed proteins harbor an array of bioactive compounds with potential health-promoting effects [15,16,17,18]. These compounds, ranging from polyphenols to phytosterols, contribute to the holistic health benefits associated with consuming these plant-based protein sources [16,19].

However, despite their nutritional virtues, hemp and rapeseed proteins also contain anti-nutritional factors that may impede optimal nutrient absorption and utilization. Understanding the presence and impact of these anti-nutritional factors is crucial for mitigating their effects and maximizing the nutritional benefits of hemp and rapeseed proteins [20,21].

Moreover, the digestibility and protein quality of hemp and rapeseed proteins play pivotal roles in determining their suitability for various food applications (Figure 2). Assessing factors such as amino acid composition, protein solubility, and protein digestibility is essential for elucidating the functional attributes and potential limitations of these plant-based protein sources in food product development.

In light of these considerations, this review paper aims to provide a comprehensive analysis of hemp and rapeseed proteins as viable options for food product development. By examining their nutritional value, bioactive compounds, anti-nutritional factors, digestibility, and protein quality, this study seeks to uncover the potential of hemp and rapeseed proteins as versatile ingredients in the creation of innovative and nutritious food products. Through rigorous exploration and analysis, this research endeavors to contribute valuable insights to the burgeoning field of plant-based food science and pave the way for the development of novel and health-enhancing food formulations.

## 2. Evaluating Hemp and Rapeseed Protein Quality

Protein quality is measured using various methods that assess the amino acid composition and digestibility of the protein. Some of the most effective and commonly used methods include calculating two scores: the Protein Digestibility Corrected Amino Acid Score (PDCAAS) and/or the Digestible Indispensable Amino Acid Score (DIAAS) [22].

The PDCAAS evaluates protein quality by comparing the amino acid profile of a protein to a reference protein pattern established by the Food and Agriculture Organization (FAO) and the World Health Organization (WHO) [23]. This method takes into account both the amino acid content and digestibility of the protein, whereas the DIAAS is a newer method developed by the FAO that considers only the digestibility of the protein, specifically the digestible indispensable amino acid content [24]. It provides a more accurate assessment of protein quality compared to the PDCAAS, especially for proteins that are less well-digested. Both scores may be useful for evaluating the nutritional quality of animal or plant-based proteins and guiding food formulation and dietary recommendations. Each method has its strengths and limitations, and researchers often use multiple approaches to assess protein quality comprehensively.

As hemp’s most predominant amino acids are edestin and 2S albumin, which are two highly digestible globular proteins [25], the PDCAAS score for the industrial hemp and hemp by-products ranges between 48% for hemp seed meal and 66% for dehulled hemp seeds, respectively. Compared to casein’s 100% PDCAAS or beef with a PDCAAS of 92%, hemp protein has a relatively low digestibility. However, it has been scientifically proved that a lower digestibility of plant-based proteins, such as grains or legumes, including hemp, is strictly correlated with the presence of anti-nutritional factors, such as phytic acid and trypsin inhibitors identified in hemp seed [26,27]. In comparison with other plant-based sources of protein, hemp has a similar PDCAAS to that of black and pinto beans, as well as lentils [28].

While both hemp and rapeseed contain all nine essential amino acids, meaning they are both complete sources of protein, the specific PDCAAS values vary depending on different research based on factors such as crop, protein concentration, extraction method, processing method, and digestibility [29,30,31].

To improve hemp’s protein digestibility score, various modern treatment techniques were applied. Ultrasonication and Ph shifting were proven to have a positive effect on the hemp’s protein structure by increasing the solubility, emulsifying activity, and foam capacity [32,33,34,35,36,37,38,39], as well as increasing the amphipathic property of the proteins by altering their globular structure [38,40,41]. There is also another method named supercritical CO_2_ extraction that allows a gentle extraction of all important bioactive compounds and minimizes the use of solvents, making it a safer alternative to traditional extraction methods [42]. Moreover, in order to provide suitable amounts of all nine amino acids, hemp protein can be paired with pea, soy, oat, and/or microalgae [43,44,45]. These combinations enable formulators to develop new products with a high protein quality, chemically close to those of animal origin [46].

The nutritional quality of rapeseed protein, in terms of PDCAAS, is higher than hemp protein and it is comparable to that of soy protein (Table 1). However, the amounts of sulfur-containing amino acids, methionine, and cysteine in canola protein exceed the nutritional requirements for both children and adults, while soy protein and casein both fall short of the requirements for infant nutrition and only casein meets the requirements for higher-age categories [47]. As well as hemp protein, rapeseed protein contains anti-nutritional factors such as erucic and phytic acid, glucosinolates, and some protease inhibitors. Different extraction techniques, including heat treatment [35], ultrafiltration [36], nanofiltration [37], alkali extraction combined with membrane processing [38], and protein micellar mass processing [39], are employed to decrease the levels of these compounds, thereby enhancing the PDCAAS score.

The DIAAS places hemp in the same range as cooked kidney beans, lentils, wheat and whey. These protein sources are classified in the no-quality-claim category (DIAAS < 75). However, by associating hemp protein with rapeseed or soy, the nutritional requirements of different ages can be met. The increased DIAAS values obtained from mixtures show the potential to achieve protein nutritional efficiency with sustainable protein sources. Nutritional efficiency lies in meeting physiological requirements with a minimal intake of high-quality protein, as opposed to a higher protein intake of low-quality protein [32].

It is to be noted that, besides digestibility, the amino acid profile of cooked protein food can differ greatly from the raw form due to a potential loss of soluble protein fractions into the boiling liquid and through the formation of amino acid derivatives [48,49,50]. With an altered amino acid composition, the DIAAS value of the cooked protein, and possibly its limiting amino acid, may differ strongly. Furthermore, animal-based proteins are also subject to variations in protein quality resulting from processing.

The understanding of the DIAAS method offers an opportunity to enhance no-quality sources of plant-based proteins such as hemp, fava bean, oats, peas, and lentils with high-quality ones, such as rapeseed or soy, therefore creating mixtures with an enhanced nutritional value and improved functionality.

**Table 1 plants-13-01195-t001:** PDCAAS and DIAAS evaluation of plant-based and animal proteins.

Food Source	PDCAAS ^1^ (%)	Ref.	Food Source	DIAAS ^1^ (%)	Ref.
Casein	100	[51,52]	Milk protein concentrate	118	[52,53]
Egg white	100	[51]	Egg	101	[32]
Soy protein concentrate	99	[54]	Soy protein concentrate	91.5	[55]
Dehulled hemp seed	66	[56]	Defatted hemp hearts	45	[56]
Rapeseed protein concentrate	93	[54]	Rapeseed protein isolate, heat-treated	100–110	[57]
Soy protein isolate	92	[54]	Soy protein isolate	90	[58]
Beef	92	[54,56]	Beef	111.6	[55]
Rapeseed protein isolate	83	[54]	Rapeseed protein isolate	76–83	[42]
Hemp seed	51	[56]	Hemp seed	54	[59]
Hemp seed meal	48	[56]	Hemp seed meal	-	
Pea protein concentrate	72	[54]	Pea protein isolate	82	[58]
Kidney beans	64.8	[53]	Cooked kidney beans	58.8	[56]
Peas	50–64	[60]	Peas	64.7	[55]
Pinto beans	57–63	[56]	Cooked pinto beans	83	[61]
Rolled oats	67	[53]	Cooked rolled oats	54.2	[53]
Black beans	53	[62]	Black beans	49	[63]
Peanuts	52	[54]	Roasted peanuts	43.34	[53]
Split red lentils	53.8	[64]	Split red lentils	50	[64]
Chickpeas	52	[52]	Cooked chickpeas	67	[52]
Whole wheat	40	[56]	Wheat	40–48	[55,58]

^1^ PDCAAS—Protein Digestibility Corrected Amino Acid Score, %; DIAAS—Digestible Indispensable Amino Acid Score, %; Ref.—reference.

## 3. Amino Acid Profile

Both rapeseed and hemp proteins are considered complete proteins, meaning they provide all essential amino acids required by the human body [9,65]. However, the composition and quantity of amino acids vary between the two (Table 2).

As indicated by Sobhy Ahmed El-Sohaimy et al. [68], hemp contains a substantial quantity of edestin, which is the predominant protein type in hemp seeds, followed by a significant proportion of globulin (67–75%) and globular albumin, ranging between 25% and 37% [69].

Overall, several researchers discovered that hemp seeds contain about 181 different proteins, which makes hemp a valuable source of biologically active compounds, polyunsaturated fatty acids, enzymes, and micronutrients beneficial for the human body [68]. Moreover, some of the enzymes found in the hemp seeds can be used to produce hydrolyzed proteins that can be successfully used in nutraceuticals or as a co-ingredient in various functional foods, as potential hypotensive agents, and as antioxidants [61,62,68,70,71]. From the chemical point of view, edestin, which is a hexameric 11S protein, and easily digestible, contains significant amounts of all essential amino acids, especially sulfur amino acids and arginine [72]. Foods formulated with the addition of arginine contained in the hemp seeds have been proven to prevent or help treat cardiovascular diseases, by regulating blood pressure [73]. Despite high-quality storage proteins, hemp seeds also contain other health-promoting amino acids, such as sulfur-containing methionine and cysteine (3.5–5.9%). When it comes to valuable/important amounts of fatty acids, hempseed oil contains over 80% polyunsaturated fatty acids (PUFAs), making it a valuable source of essential fatty acids (EFAs), specifically linoleic acid (18:2 omega-6) and alpha-linolenic acid (18:3 omega-3). The omega-6 to omega-3 ratio (n6/n3) in hempseed oil typically falls between 2:1 and 3:1, which is considered optimal for human health. Additionally, hemp seed oil contains the biological metabolites gamma-linolenic acid (18:3 omega-6; ‘GLA’) and stearidonic acid (18:4 omega-3; ‘SDA’) [74].

Nevertheless, hemp seed oil comprises a substantial quantity of tocopherols and tocotrienols (ranging from 100 to 150 mg per 100 g of oil), along with phytosterols, phospholipids, carotenes, and minerals [75,76].

In summary, the well-balanced and diverse amino acid profile of hemp protein makes it an excellent and nutritionally rich choice to support human health and diet and meet the essential needs of the health-conscious modern consumers.

According to both amino acid profiles, hemp protein stands out for its well-balanced profile, is rich in essential fatty acids, and contains a higher concentration of arginine (Table 2), an amino acid with potential cardiovascular benefits. Rapeseed protein, on the other hand, has a more favorable lysine-to-arginine ratio [77].

Compared to hemp protein, the rapeseed protein profile consists of over 45 different proteins, which is much less than hemp: 20 weakly acidic, approximately 20 neutral, and 5 basic. It has a distinct profile among other plant-based sources of protein. Rapeseed proteins are known to have structural, catalytic, and storage functions. [78]. The major storage proteins contained in rapeseed seeds are the 12S globulin (cruciferin) and the 2S albumin (napin), making up more than 70% of total rapeseed proteins [79].

Napins, characterized by their small-molecular-weight albumins (15–17 kDa), exhibit solubility across a broad pH range and possess remarkable heat stability (with a denaturation temperature Td ≥ 100 °C) [76]. The hydrophobic regions of napins are concentrated on a single side of the protein, resembling a Janus particle [78].

On the other hand, cruciferins are globulins with a higher molecular weight (300 kDa) and adopt a hexamer structure composed of two trimers [79]. The hydrophobic domains of cruciferins are distributed extensively over the protein’s surface and are also nestled within the trimeric units of cruciferins [80]. Compared to napins, the hexameric structure of cruciferins is more susceptible to undergoing structural alterations and unfolding in response to changes in temperature and pH [75,76].

Oleosin is a minor protein (1–4% by weight) present in canola seed, which functions as a stabilizer at the surface of the oil bodies so that the oil remains in the form of discrete droplets in the oil seed [80]. Nevertheless, rapeseed also consists of lipid transfer proteins and other minor proteins of a non-storage nature [79].

Overall, rapeseed contains high glutamine, glutamic acid, arginine, and leucine contents and low amounts of sulfur-containing amino acids, which are relatively altered during the industrial oil-extraction process. Indeed, the amino acid composition depends on the process used for protein extraction from the canola meal residue, which is usually up to 30% of the total [81].

Although there are well-established techniques to separate the soy protein or flax-seed protein, there are a lack of well-studied extraction methods for rapeseed protein, and therefore alternative technologies and conditions are needed, due to the differences in seed chemistry and protein profile. Details of rapeseed as a protein source in the human diet, based on research data, are also rarely found.

Although not studied enough, the viability of incorporating rapeseed meal proteins into food processing is proved by the well-balanced amino acid profile. Moreover, rapeseed/canola protein outperforms lentils, beans, and hemp protein and meets the amino acid requirements recommended by the FAO, WHO, and United Nations University (UNU) for both adults and children [82].

A widely acknowledged correlation already exists between the incorporation of plant proteins into functional foods, nutraceuticals, and various natural health products, due to their amino acid profile and nutritional value, contributing to health promotion and the positive effect it has on preventing various diseases. Plant proteins play a significant role in the food industry, and among them, rapeseed and hemp are recognized as promising co-ingredients suitable for human nutrition/diet.

## 4. Conventional and Alternative Processing Techniques

Protein concentrate production typically relies on two main extraction methods: dry fractionation or wet extraction [83].

Dry fractionation involves the separation of seeds based on their physical properties [84]. In the context of protein production, this method isolates protein-rich fractions from plant sources by utilizing mechanical techniques such as grinding, milling, sieving, or air classification [85,86,87]. These methods separate the different components based on particle size, density, or other physical characteristics.

A study conducted by Laguna et al. describes a dry-fractionation method applied to rapeseed and sunflower meals. In order to mechanically separate the protein-rich fractions, ultrafine milling was applied, followed by two separation technologies based on either particle charge (electrostatic sorting—ES) or density (turbo-separation—TS) to increase the mass yield. For rapeseed meal, electrostatic sorting was found to increase the protein and phenolic contents by 50–55% and 80–100%, respectively [88].

Dry fractionation is favored for its simplicity, cost-effectiveness, and avoidance of solvent use, making it an attractive option for manufacturers who are interested in producing protein concentrates from plant sources [89]. However, it may not yield protein purity as high as that achieved by wet-extraction methods [86].

Similarly, hemp and rapeseed protein concentrates are derived from their meals through a wet-extraction process, which involves using a liquid solvent, to dissolve and separate the protein from the insoluble portion. The resulting extract is then concentrated via centrifugation, ultrafiltration [90,91] ion-exchange, pH precipitation [92,93] or comparable methods to yield a highly pure protein with minimal carbohydrates and fat.

Can Karaca reported an efficient wet-extraction method using an isoelectric and salt precipitation on canola meal. Salt-extracted proteins, therefore collected via a centrifugation process, showed enhanced solubility and interfacial activity [94]. Another tandem wet method represented by alkaline extraction plus isoelectric precipitation offered higher efficiency (80% or more) for hemp and rapeseed protein extraction [95]. Alkaline extraction followed by acid or isoelectric precipitation and micellization extraction or reverse micellization [96] are commonly used to prepare hemp and rapeseed protein isolates with high protein content [9,97]. However, these methods were already reported to have a negative impact on hemp’s sensory properties, expressed by a non-appealing greenish color [62,98,99].

Novel protein-extraction techniques offer the potential to produce protein extracts with enriched essential amino acids and improved physicochemical and functional properties [38,39,40,41,42]. However, despite these advancements, many companies continue to rely on conventional dry methods due to concerns regarding processing costs and sustainability factors.

## 5. The Anti-Nutritional Factors of Hemp and Rapeseed Proteins

Hemp, as well as rapeseed, contain a list of off-flavor, allergenic and anti-nutritional factors. These compounds directly affect the global acceptability of the oilseeds and their by-products used in human nutrition; however, the taste and chemical behavior can be improved by protein modification. Processing techniques, such as heat treatment, are often employed to mitigate these undesirable compounds [100].

Brassica oilseeds, grains and legumes such as hemp and rapeseed but also mustard, Brussels sprouts, cabbage, cauliflower, horseradish, kale and other crops contain a series of anti-nutritive elements, which can behave both beneficially or harmlessly (differently) for animal and human food use, depending on processing conditions [93]. Anti-nutritional factors mainly refer to the glucosinolates and their derived forms, isothiocyanates, naturally produced by Brassica plants as a self-defense mechanism through various metabolic processes or mechanisms in order to avoid being eaten.

However, for food product development, other anti-nutritional elements must be considered, such as low-molecular-weight phenolic compounds, polyphenolic tannins, enzyme inhibitors, phytates, and phytic acid, as they influence the nutritional and sensory characteristics of the new products [101].

The development of new products using hemp meal requires careful consideration due to the potential presence of anti-nutritional components. One primary reason for the limited large-scale manufacturing of hemp products is the inherent risk of contamination in the final product. Hemp naturally contains variable amounts of condensed tannins, trypsin inhibitors, phytic acid, and saponins, which significantly diminish protein availability either by precipitating it or by inhibiting digestive enzymes [69,102,103,104,105]. Processes such as chelation or complex formation (involving phytic acid, condensed tannins, and saponins) can further decrease the absorption of mineral elements and vitamins [25,102,103,106].

Condensed tannins found in both hemp and rapeseed are considered anti-nutritional factors due to their ability to form undesirable complexes with proteins and other macromolecules such as starch, thus reducing the nutritional value of the product [107]. However, condensed tannins are generically phenolic compounds and can have a beneficial effect on human health due to their antioxidant capacity associated with the phenolic rings presented in their structure [108].

Another important anti-nutritional factor found predominantly in nuts, legumes and seeds such as hemp is phytic acid, the stored form of phosphorus, which works in the human body as a mineral inhibitor promoting mineral deficiency by chelating bivalent minerals like calcium, iron, zinc, and copper, as well as starch, protein, and enzymes [109]. As well as condensed tannins, ingestion in low amounts may provide several beneficial effects, proven by several recent studies due to antitumor and antioxidant capacities. It is also known that phytates can interact with proteins due to the affinity of phosphate groups for cationic amino acids. This interaction can be harmful, disrupting protein digestion, but it can also protect the human body from harmful effects of specific proteins, such as oxidases and pathogenic proteins and several diseases like diabetes [110,111].

The third group of anti-nutritional factors called saponins, biologically represent surfactants with lipophilic aglycone and hydrophilic glycosyl groups. Saponins have been shown to affect the protein digestibility of hemp by binding the activity of certain metabolic catalysts such as trypsin and chymotrypsin, thus decreasing the physiological availability of nutrients and enzymes [111].

It is important to note that the presence and levels of these antinutritional factors can vary among different varieties of hemp and may be influenced by factors such as growing conditions and processing methods. Researchers and nutritionists assess these factors to understand their impact on the nutritional quality of hemp products and to develop strategies to mitigate their effects. Processing techniques such as heat treatment or fermentation can sometimes help reduce the levels of anti-nutritional factors in hemp products. Additionally, proper preparation methods, such as soaking and cooking, can contribute to minimizing the impact of these factors when incorporating hemp into diets.

The main anti-nutritional factors contained in rapeseed are glucosinolates, sinapine, sinapic acid, tannin, phytic acid, and mucilage [112].

The problem associated with Glucosinolates, which are glycosides of β-d-thioglucose, and myrosinase activity are the most serious factors influencing the quality of rapeseed oil and meal and limiting the increased utilization of these products.

Rapeseed meals that contain intact glucosinolates cannot be used as a protein source for human nutrition due to the possibility of hydrolysis during digestion or the presence of myrosinase in other foods. Recently, several procedures have been devised for the complete aqueous extraction of glucosinolates from rapeseed meal [113], ground rapeseed, and, by diffusion, from intact rapeseed [114,115,116].

Common rapeseed meal contains significant amounts of glucosinolates, with the main five types being 3-butenylglucosinolates, 4-pentenylglucosinolates, 2-hydroxy-3-butenylglucosinolates, 2-hydroxy-4-pentenylglucosinolates, and 2-allylglucosinolates. Brassica napus meal typically contains 3-butenylglucosinolates and 2-hydroxy-3-butenylglucosinolates, while Brassica rape meal includes 3-butenylglucosinolates. It is essential to limit the total glucosinolate content in rapeseed meal for animal feed to 2.5 μmol/g, as elevated levels of these antinutritional compounds can lead to physiological disorders in animals, including hemorrhagic liver. Prolonged consumption of glucosinolate-rich foods may result in liver, kidney, and thyroid-gland enlargement in humans [108].

Moreover, the bitterness associated with certain glucosinolates in rapeseed meal can impact the acceptability of the final product. Nevertheless, glucosinolates are recognized for their natural anti-microbial and anti-carcinogenic properties [117].

Sinapine, another anti-nutritional factor presented in 70–80% of rapeseed polyphenols, is an acetylcholinesterase inhibitor [118]. Sinapic acid (SA, 4-hydroxy-3,5-dimethoxy-cinnamic acid) is the predominant free phenolic acid found in rapeseed [119]. Sinapine, also known as sinapoylcholine, is a compound found in various plants, including mustard seeds, rapeseeds, and certain other cruciferous vegetables. It is a type of alkaloid and is commonly present in the seed coats of these plants. While sinapine is not typically considered highly toxic for humans, it is important to note that excessive consumption or exposure to high levels of any compound can potentially lead to adverse effects.

Erucic acid, a monounsaturated omega-9 fatty acid, mainly common in rapeseed and mustard seeds can constitute about 30–60% of the total fatty acids of natural rapeseed [120]. The toxic effect of erucic acid was studied on animals, as exposure to diets with oils containing excessive erucic acid may lead to adverse health effects for the heart as the principal target organ. The most common effect in experimental animals is myocardial lipidosis, an accumulation of lipids in heart muscle fibers that may reduce the contractile force of heart muscles. So far, nevertheless, no evidence that dietary exposure to erucic acid is correlated to myocardial lipidosis has been established yet in humans [121]. However, there are no data to prove any heart disease-associated effects associated with the human diet. Also, the crops that are used for food product development are cultivated specifically with low erucic-acid content. Although natural forms of rapeseed and mustard contain high levels of erucic acid (over 40% of total fatty acids), levels in rapeseed cultivated for food use are typically below 0.5% [122]. Erucic acid is regulated by the European Food Safety Authority (EFSA) and the US Food and Drug Administration (FDA). In 2016, the European Food Safety Authority (EFSA) proposed a lower maximum content of erucic acid in edible oils of 2% and also recommended a tolerable daily intake of 7 mg of erucic acid per kg body weight [123]. However, there exist special crops of rapeseed named high erucic acid rapeseed (HEAR), which is a specialty rapeseed, selected for its high erucic content. It has over 50% erucic acid, and is grown as a key ingredient for plastics, personal-care products, and pharmaceuticals [124]. 

It is important to mention that nowadays advancements in breeding and processing techniques have been employed to reduce the levels of these anti-nutritional factors in rapeseed and its products, making them safer for consumption. Canola oil, for example, is derived from low-erucic-acid rapeseed varieties and has a favorable nutritional profile [125].

## 6. Bioactive Compounds

The rising number of health-conscious consumers and the significant growth in modern consumers with enthusiasm for functional foods have led to an increased demand for plant-based products that provide natural remedies for various diseases [126,127]. Both hemp and rapeseed have important bioactive compounds with nutraceutical properties.

Hemp seeds are notably rich in polyphenols (mainly flavonoids, stilbenoids, and lignanamides), alkaloids, cannabinoids, and terpenoids [128]. Flavonoids represent secondary plant metabolites composed of polyphenolic compounds with antioxidant properties that directly contribute to the hemp’s color, flavor, and potential health benefits. Hemp seeds contain unique types of flavonoids that cannot be found in any other plants, such as cannflavin A, cannflavin B, and cannflavin C [129,130]. Other kinds of flavonoids contained in hemp include quercetin, apigenin, and kaempferol. Quercetin is studied due to its anti-inflammatory properties preventing chronic inflammation linked to various health conditions, including heart disease, diabetes, and certain types of cancer [131]. Several studies also suggest that quercetin may have positive effects on cardiovascular health. It may help lower blood pressure, reduce cholesterol levels, and improve overall heart function [132]. Moreover, the flavonoids present in hemp act in a synergistic action with other compounds, such as the cannabinoids and terpenes, to produce antioxidant, antidepressant, anti-inflammatory, and disease-fighting properties [133].

Cannabinoids are a class of chemical compounds found in the cannabis plant, including hemp (*Cannabis sativa*). Hemp is known for its relatively high levels of certain cannabinoids, with a particular focus on Cannabidiol (CBD) and Tetrahydrocannabinol (THC) [134]. The effect of administrating Cannabidiol is currently investigated for its use in mood disorders such as anxiety, control for chronic pain, anti-inflammatory diseases, neurodegenerative diseases such as Alzheimer’s and Parkinson’s disease, and for its antitumorigenic properties. Of note, Cannabidiol is successfully used as a medicine against the seizure disorders Lennox–Gastaut syndrome and Dravet syndrome [135]. Industrial hemp contains no more than 0.3% concentration (on a dry-weight basis) of the psychoactive compound delta-9-tetrahydrocannabinol (THC), due to applicable European and federal law [136].

It is important to note that the therapeutic effects and potential health benefits of cannabinoids are an active area of research, and more studies are needed to fully understand their mechanisms and applications.

In addition to phytocannabinoids, hemp also contains terpenes, aromatic compounds found in many varieties of plants, including hemp. They contribute to the plant’s flavor and aroma profile [137,138]. Terpenes in hemp may also have potential therapeutic effects and can work synergistically with cannabinoids in what is known as the “entourage effect”. So far, almost 120 different terpenes have been found in hemp, but the most prevalent are myrcene, pinene, limonene, and linalool [139].

Another source of bioactive compounds identified in hemp is phytosterols. They currently present high interest for pharmaceutical companies due to a wide range of beneficial actions. One of the primary uses of phytosterols is in managing cholesterol levels. By reducing LDL cholesterol levels, phytosterols contribute to overall cardiovascular health [140]. Some research suggests that phytosterols may have anti-inflammatory effects and can also modulate the immune system [141]. Of note, hemp is a nutritious plant that contains a variety of vitamins and minerals, such as Vitamin E, particularly gamma-tocopherol, Magnesium, Phosphorus, Potassium, Zinc, Iron, Cooper, Manganese, and B Vitamins, making it a valuable addition to a balanced diet [142].

Over the years, various health-promoting compounds have been identified in rapeseed, including polyphenols, phytosterols, carotenoids, and others. However, it is worth noting that the composition and quantity of bioactive compounds present in rapeseed can vary depending on several key factors, such as the crop species, growing conditions, and processing methods [143].

Phenolic acids and their derivatives, along with both soluble and insoluble tannins, constitute the primary phenolic bioactive compounds present in rapeseed. Reports indicate that rapeseed meal can contain as much as 6% tannins. Hence, utilizing hulls, post-dehulling, as a source of natural antioxidants offers a potential avenue for their practical utilization [106].

Research indicates that rapeseed possesses a higher phenolic-component content in comparison to other seeds within the oilseed category. The phenolic compounds extracted from rapeseed have proven efficacy as potent antioxidants, finding successful applications in the realms of food, cosmetics, and pharmaceuticals [144].

From the chemical point of view, there is a total of approximately 400 mg per kg of concentration of all identified phenolic compounds in rapeseed meal. Sinapic acid stands out as the predominant phenolic compound, constituting over 85% of all quantified phenolic compounds, with an average concentration of 357 ± 13 mg/kg and a range of 339–379 mg/kg [145]. Among hydroxycinnamic acid derivatives, sinapine emerges as the most abundant bioactive compound in rapeseed. Its noteworthy properties, including antitumor, neuroprotective, antioxidant, and hepatoprotective attributes, underscore its significance for promoting health [144]. Remarkably, polyphenolic compounds, which have attracted considerable interest in recent years, are thought to confer numerous health benefits by modulating metabolic disorders, potentially through interactions with the gut microbiota [140,141]. Among the most active antioxidant components discovered in the polar fraction of rapeseed extracts is canolol (4-vinylsyringol or 2,6-dimethoxy-4-vinylphenol). Canolol is formed through the decarboxylation of sinapic acid during the roasting process of rapeseed [146].

Furthermore, nutraceutical companies are keen on substituting synthetic antioxidants like butylated hydroxyanisole (BHA), butylated hydroxytoluene (BHT), propyl gallate (PG), and tert-butylhydroquinone (TBHQ) with natural plant alternatives, as the synthetic counterparts are carcinogenic. Several previous studies have illustrated the fact that the unsaturated fatty acids and phytosterols present in rapeseed oil can effectively reduce both total cholesterol and “bad” cholesterol (such as low-density lipoprotein, LDL), while simultaneously maintaining the levels of “good” cholesterol (e.g., high-density lipoprotein, HDL). This dual action not only decreases the susceptibility to cardiovascular diseases but also acts as a preventive measure against clotting and the proliferation of vascular smooth muscle. The positive outcomes are often attributed to the optimal ratio of omega-6 to omega-3 polyunsaturated fatty acids (PUFAs), which is approximately 2:1, as supported by research studies [147,148].

Beyond their lipid solubility, these micronutrients showcase a broad spectrum of biological properties, encompassing antioxidant, anti-inflammatory, and anticancer activities [147,148].

Omega-6 fatty acids, particularly linoleic acid and its derivatives such as γ-linolenic acid, are well-known for their beneficial effects on health and are abundant in rapeseed oil. Research indicates that a diet rich in γ-linolenic acid can effectively reduce elevated levels of blood lipids, lower high blood pressure, and regulate skin perspiration [149,150]. Moreover, γ-linolenic acid demonstrates various physiological effects including anti-cancer properties, anti-thrombotic effects on the cardio-cerebrovascular system, and advantages in managing diabetes [151,152]. Conversely, α-linolenic acid is linked to physiological functions such as anti-atherosclerotic effects, facilitation of weight loss, reduction in blood lipid levels, and prevention of cardiovascular and cerebrovascular diseases [153]. The distinctive fatty acid composition found in rapeseed oil contributes to a spectrum of biological functions that support human health [143].

Carotenoids and Vitamin E present in rapeseed protein and meals also have health-promoting benefits. The main type of carotenoid found in rapeseed is (all-E)-lutein; however, there is also a minor amount of (all-E)-zeaxanthin found in cold-pressed rapeseed oil [154]. Lutein and zeaxanthin work by protecting the retina of the eye from the effects of aging. These carotenoids may prevent macular degeneration [155]. Clinical studies have also shown that lutein’s anti-oxidative and anti-inflammatory properties provide benefits by protecting and alleviating other ocular diseases like cataracts, diabetic retinopathy, myopia, and retinopathy of prematurity [156]. Rapeseed oil contains relatively high levels of tocopherols (Vitamin E) and moderate levels of vitamin K compared to other plant oils [143,157].

Overall, these bioactive compounds collectively contribute to the nutritional value and health-promoting properties of rapeseed and its derived products. Research suggests that the bioactive components in rapeseed, such as phytosterols with cholesterol-lowering effects and omega fatty acids with cardiovascular benefits, may have positive implications for human health. Furthermore, the antioxidant, anti-inflammatory, and potentially anticancer properties associated with polyphenolic compounds in rapeseed oil add to its overall health-promoting profile. As our understanding of these bioactive compounds continues to grow, incorporating rapeseed and its oil into a balanced diet may offer a range of potential health benefits.

## 7. Food Applications

The use of industrial hemp in the human diet has been a subject of debate for decades, primarily due to its chemical composition and the presence of cannabinoids, known for their potential psychotropic effects in larger quantities. In the United States, the Agriculture Improvement Act of 2018, commonly known as the 2018 Farm Bill, legalized the cultivation of industrial hemp. According to the bill, industrial hemp is defined as cannabis sativa plants containing 0.3% THC or less on a dry-weight basis [158]. Similarly, in the European Union, regulations regarding industrial hemp and THC content are established by individual member states; however, the THC levels mustnot exceed 0.3% [159,160]. Compliance with this THC threshold is crucial for hemp cultivation and the production of hemp-derived products.

Hemp protein, meal, and seed-based products (Figure 3) are readily available in various combinations on the global market. Thanks to its versatility, hemp can be consumed raw as hemp hearts (seeds) or in a processed form as powder, meal, oil, and flour. However, the most significant impact lies in serving hemp as a co-ingredient in various food products, elevating their nutritional value and asserting a compelling functional potential [19,161,162]. For the functional food market hemp stands out as an exceptionally fitting food ingredient. This is attributed to its noteworthy nutritional content and the array of health benefits it offers. Nowadays, manufacturers use industrial hemp in yogurt, snack bars, cookies, bread, pasta, milk, butter, ice cream, Beyond Meat, tofu, and other innovative/functionally improved products [65].

Seed processing techniques such as germination have been reported to promote changes in the phytochemical profile of seeds and have drawn interest to the commercial development of sprouts enriched in specific phytochemicals [163].

A study reported an increased polyphenol, flavonoid content, antioxidant activity, and protein concentration of hemp sprouts produced under a blue (B) light-led emitting diode compared to raw seed [164]. Additionally, it has been proven that hemp sprouts possess no hallucinogenic effects, do not contain high delta-9-tetrahydrocannabinol, and thus can be safely consumed without any concerns of a negative health impact [165].

Hemp seed can be also processed and manufactured into high-moisture meat analogs and hemp milk.

A high-quality and nutritious hemp milk was developed from seeds. Such milk contains about 25–30% protein and 35% fatty acids, with an optimum essential omega-3 and omega-6 fatty acid content [166,167].

Collectively, these findings demonstrate that hemp can be an excellent nutritional source of important bioactive compounds and can be successfully used for new food product development, based on accepted knowledge and new experiments.

As with hemp, rapeseed, available in various forms on the market, such as protein powder, meal, cake, flour or oil (Figure 3) may also be incorporated into some food products, particularly those designed to boost nutritional content [168].

Rapeseed has been consumed by humans as a condiment for about 3000 years. The original use of rapeseed-mustard was to mask the taste of degraded perishables. The spiciness of rapeseed-mustard is caused by a group of compounds called isothiocyanates [19].

The most used rapeseed product is rapeseed oil, also known as canola oil, a popular choice worldwide for cooking oils due to its neutral flavor, high smoke point, and healthy fat profile. It is commonly used for frying, sautéing, baking, and salad dressings.

Rapeseed oil is also used as a key ingredient in the production of margarine and spreads, providing a plant-based alternative to butter [169]. Another plant-based alternative based on rapeseed oil is mayonnaise, as rapeseed oil contributes to its creamy texture and rich flavor. When it comes to the preparation of salad dressing, marinades, and sauces, adding rapeseed oil gives a smooth texture and mild flavor.

Rapeseed cake, protein, and oil are gaining popularity as co-ingredients for baking goods, such as cakes, bread, biscuits, and gluten-free pasta. The use of rapeseed-based products is proven to enhance moisture retention and texture, extending the shelf life of the product but also improving the nutritional value of the final products [170].

Another study suggests that rapeseed flour, made from ground rapeseed meal, can be used as a gluten-free alternative in baking and cooking. Functional foods made with rapeseed flour are suitable for individuals with celiac disease or gluten intolerance. Several patents are proposed to patent the use of rapeseed protein for the production of diary-free alternatives such as plant-based milk, yogurts, and cheeses, to provide protein and improve mouthfeel [171].

With a biorefinery approach, rapeseed proteins may be extracted and recovered for high-end uses to fully exploit their nutritional and functional properties [172]. Several studies have been carried out in recent years on the applications of rapeseed/canola proteins in food products as the partial or total replacement of animal proteins. Depending on the food application tested, rapeseed proteins have been proposed as a thickener ingredient or as an emulsifier, binder, foaming, or gelling agent able to modify texture or simply fortify the protein content of a product. The range of possible food applications for rapeseed/canola proteins includes bakery and dairy products, meat, confectionery, and beverages, as well as dressings, sauces, snacks, or flavorings [172,173,174,175].

Meat alternatives incorporating rapeseed protein have been deemed acceptable based on their nutritional composition, and deemed sufficient to fulfill human requirements for essential amino acids [176]. However, the taste of rapeseed protein is reported to be unfavorable due to the presence of free and esterified phenolic acids [81]. Nonetheless, enhancements in taste can be achieved by steaming the protein concentrate and incorporating it into sausage formulations, resulting in improved taste, texture, and a distinctive aroma, thus rendering it comparable to soy-based alternatives [171,177,178].

Another study highlights the potential of using rapeseed protein isolate as a food supplement for elderly people with mastication and dysphagia problems. The obtained results show that the texture modification of food combined with rapeseed protein isolate supplementation may have a positive impact on the nutritional and sensory profile of the products [172].

Thus, hemp and rapeseed proteins offer a versatile and nutritious option for food product development, contributing to a wide range of culinary applications and meeting the needs of consumers seeking plant-based alternatives with enhanced health benefits.

## 8. Conclusions

Both rapeseed and hemp protein offer promising protein quality, with hemp showcasing a rich content of edestin and rapeseed demonstrating a high PCDAAS score. Further research is needed to comprehensively evaluate their protein digestibility and amino acid bioavailability in a specific functional food product, applying various processing methods and treatments that will meet, in the final instance, the market needs.

The presence of anti-nutritional factors in rapeseed and hemp, such as condensed tannins, trypsin inhibitors, and phytic acid, underscores the importance of careful processing techniques to mitigate their effects. Advances in extraction and processing methods offer opportunities to minimize these compounds while retaining nutritional integrity.

Both rapeseed and hemp contain bioactive compounds with potential health benefits, including polyphenols, phytosterols, and omega fatty acids. Harnessing these compounds in food formulations could contribute to functional foods targeting specific health outcomes, such as cardiovascular health and inflammation management.

Rapeseed and hemp proteins hold promise for a wide range of food applications, including plant-based meat substitutes, dairy alternatives, and baked goods. Their functional properties, such as emulsification and foaming capabilities, make them suitable for various formulations, catering to the growing demand for alternative protein sources.

The future perspective of utilizing rapeseed and hemp proteins in food product development hinges on continued research and innovation. Addressing challenges related to taste, texture, and consumer acceptance will be crucial for mainstream adoption. Additionally, exploring novel processing techniques and genetic modifications could unlock further potential for enhancing nutritional quality and functionality.

In conclusion, rapeseed and hemp proteins offer exciting opportunities for the development of innovative and sustainable food products. With careful consideration of protein quality, anti-nutritional factors, bioactive compounds, and food applications, these plant-based protein sources hold immense promise for meeting the evolving demands of the modern food industry and promoting human health and well-being.

## Figures and Tables

**Figure 1 plants-13-01195-f001:**
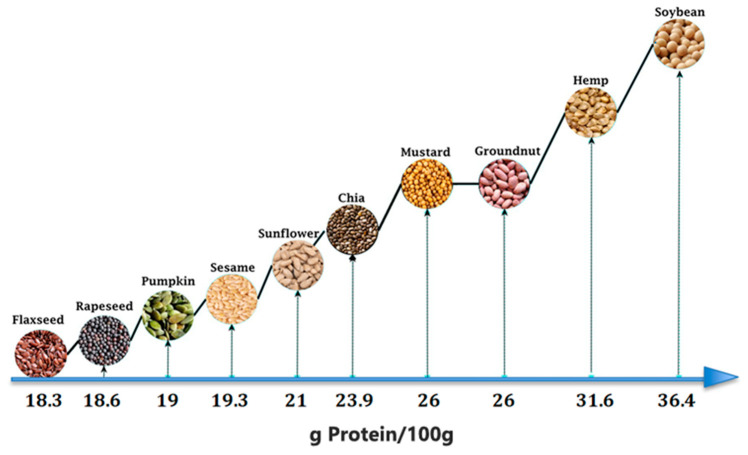
Food-use oilseed protein content per 100 g.

**Figure 2 plants-13-01195-f002:**
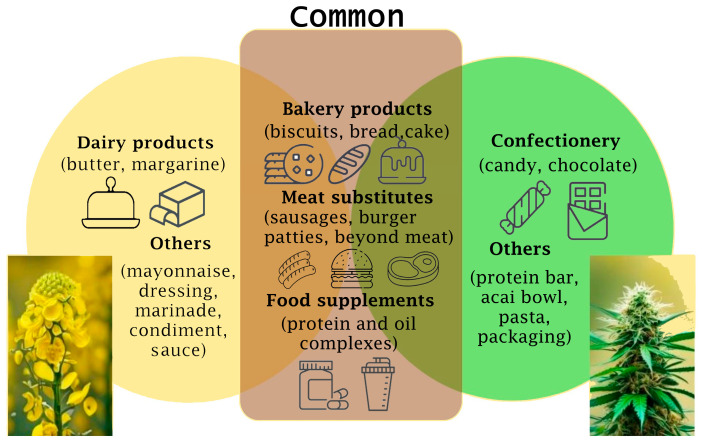
Hemp and rapeseed food applications.

**Figure 3 plants-13-01195-f003:**
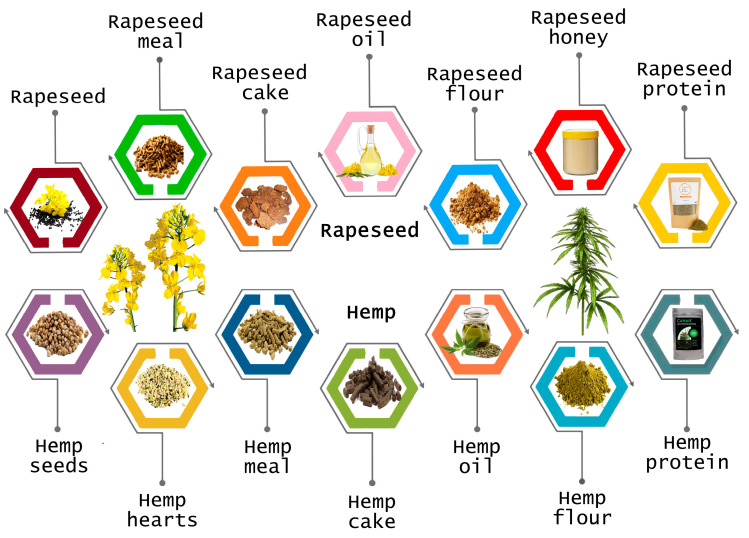
Hemp and rapeseed by-products used in the food industry.

**Table 2 plants-13-01195-t002:** Amino acid profile of hemp and rapeseed meals [66,67].

Amino Acid	Content (g/100g)	
	Hemp Meal	Rapeseed Meal
Arginine	5.56	2.02
Histidine	4.40	0.93
Isoleucine	1.79	1.41
Leucine	3.11	2.42
Lysine	5.92	1.94
Methionine	1.15	0.67
Phenylalanine	2.11	1.37
Threonine	1.68	1.45
Trypthofan	0.58	0.42
Valine	2.28	1.84
Non-essential amino acid		
Alanine	2.07	1.51
Aspartic acid	6.64	2.51
Cysteine	0.80	0.76
Glutamic acid	8.44	5.29
Glycine	2.15	1.75
Proline	1.94	1.97
Serine	2.44	1.21
Tyrosine	1.54	0.97

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
