# Peer review of "Exploring the Nutritional Potential and Functionality of Hemp and Rapeseed Proteins: A Review on Unveiling Anti-Nutritional Factors, Bioactive Compounds, and Functional Attributes"

_plants, 2024, doi:10.3390/plants13091195_

Round 1
Reviewer 1 Report
Comments and Suggestions for Authors
The manuscript of the review paper plants-2905711 is dedicated to the summarization of recent achievements in the understanding of the nutritional value and notable health benefits of hemp and rapeseed proteins. The topic is very interesting.
However, despite the fact that these studies are of special interest to scientific society, I regret to inform you that I cannot recommend the manuscript be published. The main reason is the quality of the literature research. Only 40% of references can be considered recent. Many of them are older than 20 years. I also observed many technical errors, which means that authors must be more attentive and consistent in the preparation of manuscript:
- Numerous typing mistakes: for example, lines 54, 129, 150, 409, and others.
- Mistakes in the reference citing in the text: lines 46, 51, 60, 66, 110, 120, and many others.
- Abbreviations used without explanation (lines 52, 89, 90, and others).
- What is the source of the data in Table 1?
- Reference format: for example, [1], [2], [3], [15], [36], [51], [62], [63], and others.
Moderate editing of the English language is required.
Author Response
13 April 2024
Dear Referee,
We would like to thank the referee for the close reading and for the proper suggestions. We hope that we provide all the answers to the reviewer’s comments.
Thank you very much for the recommendations which help us to improve our paper entitled “Exploring the Nutritional Potential and Functionality of Hemp and Rapeseed Proteins: A review on unveiling Anti-Nutritional Factors, Bioactive Compounds, and Functional Attributes”.
The present version of the paper has been revised according to the reviewer’s suggestions.
We uploaded the corrected version of the article for which we used the yellow color for the addition text.
Reviewer comments
Reviewer: The manuscript of the review paper plants-2905711 is dedicated to the summarization of recent achievements in the understanding of the nutritional value and notable health benefits of hemp and rapeseed proteins. The topic is very interesting.
Response: We want to thank to the referee for the close reading of our manuscript. Indeed the topic is interesting and we want to futher develop food products with hemp and rapeseed proteins.
Reviewer: However, despite the fact that these studies are of special interest to scientific society, I regret to inform you that I cannot recommend the manuscript be published. The main reason is the quality of the literature research. Only 40% of references can be considered recent. Many of them are older than 20 years. I also observed many technical errors, which means that authors must be more attentive and consistent in the preparation of manuscript:
- Numerous typing mistakes: for example, lines 54, 129, 150, 409, and others.
- Mistakes in the reference citing in the text: lines 46, 51, 60, 66, 110, 120, and many others.
- Abbreviations used without explanation (lines 52, 89, 90, and others).
- What is the source of the data in Table 1?
- Reference format: for example, [1], [2], [3], [15], [36], [51], [62], [63], and others.
Response: Thank you for your time spent to review this manuscript and for taking into consideration our topic. We appreciate your valuable feedback and constructive criticism. We understand and acknowledge your concern regarding the quality of the literature research. Therefore we diligently revised the whole reference list, added more recent information to ensure a higher proportion of recent references and make the manuscript more relevant/informative. Additionally, as you suggested we read thoroughly the manuscript to rectify the technical errors and paid more attention to detail during this process. We have also revised the publishing rules that Plants journal has, to make sure the references are cited correctly. We used recently references, completed the sources for Table 1, we explained the abbreviations, the references format and we corrected the typing mistakes. Also, all the article have been revised by an English teacher.
Sincerely,
Georgiana CODINĂ and Marina AXENTII
Reviewer 2 Report
Comments and Suggestions for Authors
The authors comprehensively reviewed hemp and rapeseed proteins' nutritional potential and functionality. Anti-nutritional factors, bioactive compounds, and potential food applications of investigated proteins are discussed. The study is informative. However, numerous reports regarding this topic have been published thus far, so this study lacks a novelty. The authors are encouraged to straighten their manuscript by identifying economic barriers and research gaps that need to be addressed to mitigate the disadvantages of processing techniques used so far. A subsection dealing with processing techniques should be added. In addition, the lack of adequate references makes reviewing the manuscript difficult.
Specific comments:
- Lines 29-34. Please provide references for the statements.
- Abbreviations should be expanded when first mentioned in the text (e.g., USDA)
- Lines 171-172. Please provide the reference.
- Lines 178-180. Add references.
- Lines 203-206. Add references.
- Lines 335-341. Add references.
- Lines 343-348. Add references.
- Please correct the statement in Lines 349-350. As indicated in [77]-"In 2016, the European Food Safety Authority (EFSA) proposed a lower maximum content of erucic acid in edible oils of 2% (instead of 5%) and also suggested a tolerable daily intake of 7 mg erucic acid per kg body weight.".
- Lines 354-358. Add references.
- Lines 386-388. Add references.
- Lines 408-412. Add references.
- Lines 422-427. Add references.
- Line 435. Inadequate references.
- Lines 436-439. Add references.
- Please check the references throughout the manuscript.
- Lines 463-464. The authors stated, "Main type of carotenoid found in rapeseed is (all-E)-lutein, but studied carried by blab la also determined in cold pressed rapeseed oil a minor amount of 464 (all-E)-zeaxanthin". What is blab la?
- Line 489. Check whether the reference [105] is adequate.
- Check the format of the References. Some are incomplete.
Comments on the Quality of English Language
The manuscript needs extensive English editing.
Author Response
13 April 2024
Dear Referee,
We would like to thank the referee for the close reading and for the proper suggestions. We hope that we provide all the answers to the reviewer’s comments.
Thank you very much for the recommendations which help us to improve our paper entitled “Exploring the Nutritional Potential and Functionality of Hemp and Rapeseed Proteins: A review on unveiling Anti-Nutritional Factors, Bioactive Compounds, and Functional Attributes”.
The present version of the paper has been revised according to the reviewer’s suggestions.
We uploaded the corrected version of the article for which we used the yellow color for the addition text.
Reviewer comments: The authors comprehensively reviewed hemp and rapeseed proteins' nutritional potential and functionality. Anti-nutritional factors, bioactive compounds, and potential food applications of investigated proteins are discussed. The study is informative. However, numerous reports regarding this topic have been published thus far, so this study lacks a novelty. The authors are encouraged to straighten their manuscript by identifying economic barriers and research gaps that need to be addressed to mitigate the disadvantages of processing techniques used so far. A subsection dealing with processing techniques should be added. In addition, the lack of adequate references makes reviewing the manuscript difficult.
Response: We want to thank to the referee for the close reading of our manuscript and his/er suggestions. We completed our manuscript with new references in order to improve it. Also, according to referee suggestions we completed our manuscript with a subsection with processing techniques used for both hemp an rapeseed protein extraction in order to discuss the conventional and alternative methods used by other researchers in their studies and point out the technological advancements made in this field of study.
Reviewer comments: Specific comments:
- Lines 29-34. Please provide references for the statements.
Response: We provided.
- Abbreviations should be expanded when first mentioned in the text (e.g., USDA)
Response: We revised.
- Lines 171-172. Please provide the reference.
Response: We provided.
- Lines 178-180. Add references.
Response: We provided.
- Lines 203-206. Add references.
Response: We provided.
- Lines 335-341. Add references.
Response: We provided.
- Lines 343-348. Add references.
Response: We provided.
- Please correct the statement in Lines 349-350. As indicated in [77]-"In 2016, the European Food Safety Authority (EFSA) proposed a lower maximum content of erucic acid in edible oils of 2% (instead of 5%) and also suggested a tolerable daily intake of 7 mg erucic acid per kg body weight.".
Response: We corrected.
- Lines 354-358. Add references.
Response: We provided.
- Lines 386-388. Add references.
Response: We provided.
- Lines 408-412. Add references.
Response: We provided.
- Lines 422-427. Add references.
Response: We provided.
- Line 435. Inadequate references.
Response: We revised.
- Lines 436-439. Add references.
Response: We provided.
- Please check the references throughout the manuscript.
Response: We checked once again the references throughout the manuscript.
- Lines 463-464. The authors stated, "Main type of carotenoid found in rapeseed is (all-E)-lutein, but studied carried by blab la also determined in cold pressed rapeseed oil a minor amount of 464 (all-E)-zeaxanthin". What is blab la?
Response: We revised.
- Line 489. Check whether the reference [105] is adequate.
Response: We checked
- Check the format of the References. Some are incomplete.
Response: We checked once again the references throughout the manuscript.
Also, all the manuscript have been revised by an English teacher.
Sincerely,
Georgiana CODINĂ and Marina AXENTII
Reviewer 3 Report
Comments and Suggestions for Authors
*First of all, the writing rules of the article should be revised and corrected according to the writing rules of Plants.
* The topics explained in the study should be combined in the same paragraph by addressing the following topics one by one, and they should not be scattered.
*Information about bioactive compounds and amino acids should be given in tabular form rather than verbal presentation.,
*Numerical values related to the subjects should be presented in tables or graphics for a better understanding.
if below reference is added to text, better:
-Some rape/canola seed oils: fatty acid composition and tocopherols
Comments on the Quality of English Languageshould be moderate editing,please.
Author Response
13 April 2024
Dear Referee,
We would like to thank the referee for the close reading and for the proper suggestions. We hope that we provide all the answers to the reviewer’s comments.
Thank you very much for the recommendations which help us to improve our paper entitled “Exploring the Nutritional Potential and Functionality of Hemp and Rapeseed Proteins: A review on unveiling Anti-Nutritional Factors, Bioactive Compounds, and Functional Attributes”.
The present version of the paper has been revised according to the reviewer’s suggestions.
We uploaded the corrected version of the article for which we used the yellow color for the addition text.
Reviewer comments: First of all, the writing rules of the article should be revised and corrected according to the writing rules of Plants.
Response: We want to thank the referee for the close reading of our manuscript. We revised once again the article in order to be written according to the Plants rules.
Reviewer comments: The topics explained in the study should be combined in the same paragraph by addressing the following topics one by one, and they should not be scattered.
Response: We want to thank the referee for his/her recommendation. We tried to improve more our manuscript. We hope to be finer now.
Reviewer comments: Information about bioactive compounds and amino acids should be given in tabular form rather than verbal presentation. Numerical values related to the subjects should be presented in tables or graphics for a better understanding.
Response: We agree on referee recommendation and want to thank to him/her for the suggestions. We revised our manuscript and according to the referee’s suggestions we presented information about amino acids in tabular form to facilitate a more organized and concise presentation of this data. However, the bioactive compounds contained in both sources are hard for us to be included in a table, as they both have common and distinct bioactive compounds with suitable health benefits and each source has its own set of proven and distinct health benefits.
Reviewer comments: if below reference is added to text, better:
-Some rape/canola seed oils: fatty acid composition and tocopherols
Response: We completed the manuscript with this reference.
Also, all the article have been revised by an English teacher.
Sincerely,
Georgiana CODINĂ and Marina AXENTII
Round 2
Reviewer 1 Report
Comments and Suggestions for Authors
I am satisfied with most of the corrections. However, I still have some minor comments:
1. Please check the use of abbreviations.
PDCAAS was once explaned in lines 90–91, and it is not necessary to repeat it in lines 92, 112, and 129.
same for DIAAS: lines 91, 96, and 140; FAO: lines 92 and 243; WHO: lines 94-95 and 243-244.
2. Please check the format of reference 165.
3. Table 1 can be the width of the main text.
Comments on the Quality of English LanguageMinor editing of the English language is required.
Author Response
Dear Referee,
We would like to thank the referee for the close reading and for the proper suggestions. We hope that we provide all the answers to the reviewer’s comments.
Thank you very much for the recommendations which help us to improve our paper entitled “Exploring the Nutritional Potential and Functionality of Hemp and Rapeseed Proteins: A review on unveiling Anti-Nutritional Factors, Bioactive Compounds, and Functional Attributes”.
The present version of the paper has been revised according to the reviewer’s suggestions.
We uploaded the corrected version of the article for which we used the yellow/red color for the addition text.
Reviewer comments
Reviewer: I am satisfied with most of the corrections. However, I still have some minor comments:
Response: We want to thank the referee for the close reading of our manuscript. Thank you very much for your appreciation.
Reviewer: 1. Please check the use of abbreviations.
PDCAAS was once explaned in lines 90–91, and it is not necessary to repeat it in lines 92, 112, and 129.
same for DIAAS: lines 91, 96, and 140; FAO: lines 92 and 243; WHO: lines 94-95 and 243-244.
Response: Thank want to thank to the reference for the close reading of our manuscript. We have corrected all of them, as you suggested.
Reviewer: 2. Please check the format of reference 165.
Response: The format of reference number 165 has been updated. Thank you!
Reviewer: 3. Table 1 can be the width of the main text.
Response: As you mentioned, we have adjusted the width of the table 1.
Sincerely,
Georgiana CODINĂ and Marina AXENTII
Reviewer 2 Report
Comments and Suggestions for Authors
The authors improved their work.
-There is no need to repeat the extensions of abbreviations several times in the text. The abbreviations (e.g., PDCAAS and DIAAS, FAO, etc.) should be expended only when first mentioned in the text. Afterward, only abbreviations should be used in the manuscript.
- Lines 475-476. The authors did not change the references as suggested by the reviewer. In references 141, 143, and 145, which the authors introduced, canolol is not even mentioned. Please insert adequate references. The authors should check every reference in the manuscript to present their work correctly.
- Line 396. "It's important to mentioned" should be corrected into "It's important to mention ". Please check the grammar throughout the manuscript.
Comments on the Quality of English LanguagePlease check the grammar throughout the manuscript.
Author Response
Dear Referee,
We would like to thank the referee for the close reading and for the proper suggestions. We hope that we provide all the answers to the reviewer’s comments.
Thank you very much for the recommendations which help us to improve our paper entitled “Exploring the Nutritional Potential and Functionality of Hemp and Rapeseed Proteins: A review on unveiling Anti-Nutritional Factors, Bioactive Compounds, and Functional Attributes”.
The present version of the paper has been revised according to the reviewer’s suggestions.
We uploaded the corrected version of the article for which we used the yellow/red color for the addition text.
Reviewer comments
Reviewer: The authors improved their work.
Response: We want to thank the referee for the close reading of our manuscript. Thank you very much for your appreciation.
Reviewer: -There is no need to repeat the extensions of abbreviations several times in the text. The abbreviations (e.g., PDCAAS and DIAAS, FAO, etc.) should be expended only when first mentioned in the text. Afterward, only abbreviations should be used in the manuscript.
Response: We want to thank the referee for the close reading of our manuscript. We revised. As you suggested, the abbreviations were explained in text only once and were used afterward accordingly.
Reviewer: - Lines 475-476. The authors did not change the references as suggested by the reviewer. In references 141, 143, and 145, which the authors introduced, canolol is not even mentioned. Please insert adequate references. The authors should check every reference in the manuscript to present their work correctly.
Response: We want to thank the referee for the close reading of our manuscript. We revised the references, we checked them one more time and we inserted adequate references.
Reviewer: - Line 396. "It's important to mentioned" should be corrected into "It's important to mention ". Please check the grammar throughout the manuscript.
Response: We want to thank the referee for the close reading of our manuscript. We corrected and once again the manuscript has been revised by an English teacher.
Sincerely,
Georgiana CODINĂ and Marina AXENTII
Reviewer 3 Report
Comments and Suggestions for Authors
I check revised version of article. Allcorrections were made by authors
Comments on the Quality of English Languagesuitable.
Author Response
Dear Referee,
We would like to thank the referee for the close reading and for the proper suggestions. We hope that we provide all the answers to the reviewer’s comments.
Thank you very much for the recommendations which help us to improve our paper entitled “Exploring the Nutritional Potential and Functionality of Hemp and Rapeseed Proteins: A review on unveiling Anti-Nutritional Factors, Bioactive Compounds, and Functional Attributes”.
The present version of the paper has been revised according to the reviewer’s suggestions.
We uploaded the corrected version of the article for which we used the yellow/red color for the addition text.
Reviewer comments
Reviewer: I check revised version of article. All corrections were made by authors
Response: We want to thank the referee for the close reading of our manuscript. Thank you very much for your appreciation. Our article have been revised one more time by an English teacher.
Sincerely,
Georgiana CODINĂ and Marina AXENTII